# Novel Reinforcing Techniques and Bearing Capacity Analysis for Tunnel Lining Structures with Extensive Corrosion

**DOI:** 10.3390/ma16072871

**Published:** 2023-04-04

**Authors:** Xicao Zha, Mingfeng Lei, Ningxin Sun, Yongheng Li, Linghui Liu, Lian Duan, Lichuan Wang

**Affiliations:** 1School of Civil Engineering, Central South University, Changsha 410075, China; 8210201413@csu.edu.cn (X.Z.);; 2MOE Key Laboratory of Engineering Structures of Heavy Haul Railway (Central South University), Changsha 410075, China; 3Shandong Provincial Communications Planning and Design Institute Group Co., Ltd., Jinan 250031, China; 4Suzhou Municipal Construction Management Center, Suzhou 215000, China; 5Zhangjiajie Works Section, China Railway Guangzhou Bureau Group Co., Ltd., Zhangjiajie 427000, China; 6China Railway 18th Bureau Group Co., Ltd., Tianjin 300222, China

**Keywords:** tunnel engineering, lining corrosion, lining reinforcement, load-bearing capacity calculation lining structure reinforcement design method

## Abstract

Affected by the erosive environment, tunnel lining concrete in the long-term service zprocess often exhibits engineering diseases such as concrete corrosion degradation and loss of strength, decreasing the stability of the tunnel lining structure and the traffic safety. Based on HTG tunnel project, the basic distribution rule of tunnel lining corrosion and macro mechanical properties of corroded concrete were explored in this paper through engineering disease site investigation. Then, on this basis, aiming at large-scale corrosion of tunnel lining structure, two reinforcement and repair schemes are proposed, corrugated steel plate reinforcement method and channel steel reinforcement method. Indoor component tests are carried out on the two reinforcement schemes. The failure characteristics and stress and deformation law of tunnel lining members after reinforcement and repair were verified. The analysis showed that the failure process of the reinforced specimens on the tensile side could be divided into the non-cracking stage and the working stage with cracks, and the cracking load and failure load of the specimens were significantly increased. The bearing capacity of the reinforced specimens was divided into the ultimate bearing capacity against cracking and the ultimate bearing capacity during failure. Finally, the calculation methods of the bearing capacity of the channel steel reinforcement method and the corrugated steel plate reinforcement method were derived. Comparative analysis shows that the results of numerical simulation, experimental testing and theoretical simplification methods are close to each other, and the maximum deviation is less than 8%. The established method for calculating the bearing capacity of corroded components after reinforcement is reliable and can be used for the design calculation of corroded lining reinforcement.

## 1. Introduction

By the end of 2020, there were 21,316 road tunnels (15,285 km) and 16,798 railroad tunnels (19,630 km) built in China [1], of which more than 4000 old railroad tunnels have been in service for more than 40 years [2]. Subject to early construction quality as well as long-term environmental erosion and dynamic train loading, these old tunnels commonly suffer from water leakage, lining cracking, concrete corrosion and other disease problems, which seriously affect the normal operation of the line [3,4,5]. In particular, in the western region of China, affected by the aggressive sulfate environment, the tunnel lining concrete corrosion generally existed, such as: the Tanjiazhai highway tunnel lining side walls in Hubei Province, sulfate corrosion is serious and accompanied by water seepage recrystallization phenomenon [5,6]; as for more than a dozen operating tunnels on the Chengkun Railway, their lining concrete is deteriorated to swelling, non-caking and collapse-like under the combined erosion of sulfate, temperature and humidity environment [7]. The Shiziya tunnel in Hubei was forced to be closed for maintenance due to local loss of strength of tunnel concrete only two years after opening [8]. After grouting and other measures were adopted to control the tunnel in Liupanshan, Ningxia, the corrosion and deterioration continued to occur, and large cracks and concrete spalling appeared at the sidewalls and vault of the tunnel, endangering traffic safety [9]. Therefore, how to reinforce and repair these corroded and deteriorated operating tunnels has become a major problem and urgent need in the field of tunnel engineering.

For the reinforcement and repair technology of corrosion-deteriorated tunnel lining structure, Britain, the United States and other countries have performed relevant studies earlier, and proposed many practical local reinforcement and reinforcement schemes. For example, Asakura et al. [10] used high-strength carbon fiber material to reinforce and repair the corroded lining, which effectively improved the bearing capacity of the cross-section. With the continuous increase in the number of old tunnels in recent years, some Chinese experts and scholars have also carried out corresponding studies [11,12]. Li et al. [13] established a 1:5 scaled-down model with a damaged tunnel as a prototype, and comprehensively analyzed the load-bearing performance and reinforcement effect of three types of arch-set reinforcement schemes, RC, ECC and R/ECC. The results show that the lining reinforced by R/ECC arch set has better structural bearing capacity and deformation performance, and shows excellent reinforcement effect in various damage states of lining reinforcement. In conclusion, many scholars have carried out more research on tunnel structure reinforcement and repair techniques, forming representative schemes such as the adhesive steel method [14,15,16,17], the paste fiber material method [18,19,20,21] and the cementitious reinforcement method [22,23,24,25], which can effectively improve the stress state of diseased tunnels and alleviate the defects and safety hazards caused by lining steel corrosion thickness reduction of lining steel.

However, for the large area corrosion deterioration of plain concrete lining structure, due to the fact that the lining concrete has lost its structural strength within a certain thickness range, it is impossible to paste fiber material or anchor steel plate on the surface of the lost strength during actual repair. Hence, it is often necessary to chisel out the severely corroded part before the next step of reinforcement construction. Meanwhile, the mechanical conversion mechanism and design method of the lining structure after reinforcement and repair are still relatively vague, In the actual construction, the repair design scheme is mostly determined by the method of empirical judgment, and the reliability and economy of the design lack of rigorous theoretical basis.

To this end, this paper relies on the HTG tunnel project, firstly, through the disease site investigation, to grasp the basic distribution law of the tunnel lining corrosion and the macro mechanical properties of corroded concrete. Further, on this basis, for the large area of corroded tunnel lining structure, two types of reinforcement and repair solutions are proposed: corrugated steel plate reinforcement method and channel steel reinforcement method. Then, indoor component tests were carried out on the two reinforcement schemes to investigate the damage characteristics, force deformation law of the lining components after reinforcement and repair. Finally, the calculation method of the bearing capacity of the corroded tunnel lining structure after reinforcement was established.

## 2. Tunnel Lining Corrosion Site Investigation

The research team went to the site several times to investigate and detect the engineering diseases of the HTG tunnel. Field investigation and analysis found that the main damage types of tunnel lining concrete are: leakage, salt corrosion damage, concrete carbonization, weathering and spalling, etc. [26].

(1) Water quality tests showed that the measured sulfur dioxide concentration in the groundwater of the HTG tunnel exit section was as high as 618.9 mg/L, and the environmental action level of the concrete structure was V-C, which is a severe chemical erosion environment [6,26,27].

(2) No anti-sulfate erosion cement was used in the actual construction, and the lining concrete showed serious corrosion deterioration under the effect of sulfate erosion for many years. Statistically, a thick layer of salt crystals attached to the concrete surface (width about 0.5–2 m) existed almost every 2–3 m within the 100 m long range of HTG tunnel exit, and the concrete on the lining surface was swollen and collapsed, with serious loss of strength [6,26,28,29], as shown in Figure 1a,b.

(3) The results of on-site drilling and coring showed that the concrete in the severely corroded area could no longer form a complete core sample after being disturbed by the drilling rig. The cementing material inside the core was seriously lost, and the concrete was densely filled with holes and soil particles brought by groundwater, the material completely lost strength, the maximum corrosion depth of about 20 cm, as shown in Figure 1c. In the mildly corroded area, the core is basically intact, but there are still many holes on the surface of the core. Reddish brown soil particles can be seen at the fracture of the core, indicating that some cementing materials are lost in the lining due to groundwater erosion, and holes are formed inside and soil particles are partially filled, as shown in Figure 1d. The measured compressive strength of concrete is 17–24 MPa, and the average value is 22.9 MPa [6,30].

## 3. Design and Test Scheme for Reinforcement of Corroded Tunnel Lining Structure

### 3.1. Reinforcement Scheme

After large-scale corrosion of tunnel lining, the strength of concrete decreases, which weakens the overall stability of tunnel lining structure [30,31]. Therefore, according to the corrosion range/depth distribution characteristics of the lining concrete obtained from the field investigation in Section 2, combined with the existing engineering experience, corrugated steel plate or channel steel method is designed to strengthen the scheme.

Channel steel reinforcement method. In this reinforcement scheme, the corroded part of the concrete lining is first chiseled away, and then the structure is reconstructed with channel steel and concrete to achieve structural replacement and strengthening. In this case, the channel steel is fixed on the uncorroded concrete by expansion bolts, and then the slightly expansion concrete [12,31] is poured into the chiseled part, as shown in Figure 2.

Corrugated steel plate reinforcement method. The program is to cut out the corroded concrete after the installation of corrugated steel plate, with a special location of rivets, anchor bolts and concrete filling between the steel plate and the existing lining structure, so as to achieve the corrosion of the inner lining structure of the application, reinforcing and strengthening the overall lining structure, as shown in Figure 3.

### 3.2. Experimental Design and Modeling

Considering the actual conditions of the relying project (only plain concrete lining) and the structural force characteristics, two types of reinforcement models were designed as shown in Figure 4.

Meanwhile, considering the test requirements, five eccentric compressive members were made for the two types of reinforcement schemes, including two reinforcement specimens by channel method, two reinforcement specimens by corrugated plate method and one plain concrete specimen (control of reinforcement effect). Specific specimen parameters are shown in Table 1.

The original concrete was poured with C30 commercial concrete, and the measured average uniaxial compressive strength of the cube was 34.1 MPa. The repaired concrete was made of slightly expansion concrete, and the mix ratio is shown in Table 2, in which the cement was ordinary silicate cement (P·O 425) produced by Changsha Southern Cement Company. The coarse aggregate was apparent density 2720 kg/m^3^, particle size 5–10 mm and the fine aggregate was natural yellow sand with fineness modulus 2.7. The admixture was polycarboxylic acid high efficiency air-entraining water-reducing agent and calcium sulfoaluminate expansion agent. The mixing water was tap water. After maintenance and testing, the average uniaxial compressive strength of the cubes was measured to be 36.3 MPa.

In addition, to ensure the connection between the reinforced steel plate and concrete, rivets or expansion bolts are set behind the reinforced steel plate, respectively, as shown in Figure 5.

### 3.3. Test Loading and Measurement Point Arrangement

As shown in Figure 6, the test is a static test, with one-way knife hinge support at the upper and lower ends of the specimen, and the stress mode of the specimen is one-way eccentric load, which is loaded by the hydraulic device at the bottom; the stress, strain or displacement values at each typical position are recorded in real time.

In Figure 6a, S1–S7 are resistive strain gauges for measuring the strain in the direction of the plane of action of the bending moment. D1–D5 (range 10 mm) are horizontal displacement gauges, which are used to measure the lateral displacement and bending curvature change of each section of the specimen.

After the specimen and its measuring instrument are installed in place, the oil delivery valve is slowly opened for preloading (about 20% of the estimated limit load), and the working state of the measuring instrument is observed to be normal. After confirming the correctness, the test piece is unloaded and cleared for formal loading. Press readings were loaded at a rate of 0.3 kN/s until 80% of the estimated ultimate load was reached, then reducing to 0.1 kN/s until complete failure of the specimen.

## 4. Test Results and Analysis

### 4.1. Deformation and Damage Characteristics of Specimens

Figure 7 shows the load–deflection test results of the specimens reinforced on the tension side and the specimens strengthened on the compression side.

From the analysis, it can be seen that:(1)The load–deflection curve of the reinforced specimen on the tensile side shows an overall two-stage evolution relationship, as shown in Figure 7a, which exhibits certain ductile damage characteristics, and the load carrying capacity of the reinforced specimen is improved significantly. Specifically:
At the initial stage (before cracking), the specimen material is in the elastic stage, and the deflection in the column increases linearly with the load until the concrete in the tensile area cracks, at which time the separation of steel plate and concrete occurs in the corrugated steel plate reinforcement method. Thereafter, the specimen enters the working stage with cracks, its cross-sectional stiffness decreases as a whole, the tensile stress is continuously provided by the channel steel or corrugated steel plate, and the concrete stress on the compressed side increases rapidly.The cracking loads of the channel method and corrugated steel plate method reinforced specimens reached 415.6 kN and 556.8 kN, respectively, which increased by 72% and 130.5% compared with the unreinforced plain concrete specimens. After the development of the working stage with cracks, the breaking loads of the channel method and corrugated steel plate method reinforced specimens were 797.2 kN and 823.7 kN, respectively. The bearing capacity was increased by 2.3 times and 2.4 times of that of the plain concrete, respectively.The deflection in the column of the specimens reinforced by the channel method and the corrugated steel plate method at the tensile side was 3.85 mm and 4.27 mm, respectively, while the deflection at the unreinforced specimen was 2.55 mm, which shows that the deformation performance of the specimens reinforced by the tensile side was also improved.
(2)The load–deflection curves of the reinforced specimens on the compressive side showed an overall linear variation relationship, as shown in Figure 7b, which exhibited significant brittle damage characteristics. Compared with the plain concrete reference specimens, the critical damage loads of the reinforced specimens on the compressed side were elevated to a limited extent, about 10%. The specific test damage processes of each specimen are:
Plain concrete specimen: when the load reached 241.6 kN, the test load dropped rapidly, and a crack with a penetration depth of 16 cm appeared on the tensile side of the specimen, and the concrete lost its load-bearing capacity by tensile cracking.Channel steel method of reinforced specimens on the compression side: before the specimen was damaged, the testing machine screen load value continued to rise, and there are no cracks on the concrete surface. When the load reached 271.3 kN, the testing machine screen load rapidly declined. Meanwhile, the test specimen tensile side presents a crack with the depth of 25 cm; the test specimen tensile zone concrete reached the ultimate bearing capacity cracking, and the concrete in the compression zone did not see crushing and spalling.Corrugated steel plate method of compression side reinforcement specimen: the damage process is similar to the channel steel method of compression side reinforcement specimen, and when the load reaches 263.8 kN, the tensile area of the specimen cracks and destroys.
(3)After the corrosion of tunnel lining, the local stiffness of the lining decreases, and stiffness changes affect the overall structure of the lining internal force distribution, which often leads to the increase in the bending moment of adjacent areas of the corroded lining, and further affects the structural safety of adjacent parts. The ratio of the load to deflection value at the moment of damage of the specimen is defined as its stiffness, that is
(1)$$EI=\frac{Nel^{2}}{\pi^{2}f}$$
where, *EI* is the flexural stiffness; *N* is the axial load of the eccentrically compressed member; *e* is the loading eccentricity of the specimen; *l* is the height of the specimen; and *f* is the lateral deflection of the central section of the specimen.

Thus, according to the experimental results, the flexural stiffness of the specimen at the peak load point of different repair schemes can be obtained, as shown in Table 3.

The analysis shows that compared with the unreinforced plain concrete specimen, the flexural stiffness of the specimen increased by more than 1 times after reinforcement by the channel method or corrugated steel plate method on the tensile side, respectively, and the reinforcement effect was significant [20,30]. However, when the channel steel or corrugated steel plate method is also used to strengthen the compressive side of the specimen, the flexural rigidity of the specimen is only increased by 25% and 11.4%. It can be seen that the improvement effect of repairing on the compressive side on the stiffness of the member is limited.

### 4.2. Cross-Sectional Strain Characteristics

Figure 8 gives the strain distribution of the cross-section in the column of each specimen, and the analysis shows that:

(1) Cracks appeared on the tensile side of the specimen after the bias load *N* reached 415.6 kN in the tensile side of the reinforced specimen by the channel method, resulting in the measured data of strain gauges S1 and S2 not truly reflecting the strain of the internal channel steel. The strain data at a height of 50 cm in the cross-section were replaced with the data measured by strain gauges S2′ affixed to the waist of the channel steel to describe the strain after the channel steel had assumed the full tensile stress. As seen from the figure, the axial strain of the section in the specimen column basically follows a linear distribution before cracking [31]. After the cracking of the specimen, except that the data of S1 cannot truly describe the strain state due to the surface cracking, the strain of other sections also basically satisfies the assumption of flat section.

(2) In the process of loading the tensile side reinforced specimen with corrugated steel plate method, the cracking load is 627.2 kN. The axial strain of the column section of the specimen before cracking is basically in line with the linear distribution, meeting the assumption of flat section. The corrugated steel plate, repaired concrete and corroded concrete work well together, and the ultimate bearing capacity is 823.7 kN.

(3) The axial strains in the column section of the three specimens, namely, the reinforced specimen in the compression side of the channel method, the reinforced specimen in the compression side of the corrugated steel plate method and the unreinforced plain concrete specimen, are basically distributed linearly and can satisfy the assumption of flat section. When the specimens reached the ultimate load, the concrete strain values at the edge of the tensile zone were all about 100 *με*, and the specimens exhibited brittle tensile damage characteristics.

## 5. Large Area Corrosion Lining Reinforcement Design Method

According to the analysis of test results in Section 4.1, it can be found that for the compression side reinforcement, no matter what reinforcement method is used, its effects on the overall structural load-bearing capacity and stiffness enhancement are limited. Therefore, this paper only carries out specific research on the design method of the tensile side reinforcement. Through several basic hypotheses, the force diagram of the two strengthening methods on the tensile side is obtained, and the cracking bearing capacity and ultimate bearing capacity of the specimens on the tensile side are calculated and deduced.

### 5.1. Basic Working Hypotheses

It can be seen from the test data in Section 4.2 that the axial strain of the section of the specimen column basically follows a linear distribution, which can satisfy the assumption of flat section. Furthermore, on this basis, to highlight the point and simplify the calculation, the following assumptions are further made in this section:

(1) The eccentricity of the specimen should be not less than 0.2 times of the height of the section.

(2) The average strain of the section varies linearly along the horizontal height direction, satisfying the assumption of flat section.

(3) The corroded concrete is well bonded to the reinforcement and repaired concrete, which can coordinate deformation and work together. For the reinforced repaired specimen, when the concrete strain at the edge of the tensile zone reaches 300 µε, the specimen concrete cracks, after which the concrete on the tensile side withdraws from the bearing work and the tensile stresses is all provided by the channel steel or corrugated steel plate.

(4) When the specimen reaches the ultimate bearing capacity, the tensile side of the channel steel or corrugated steel plate does not reach the yield strength and the edge strain of the compressive zone is 2500 µε at this time.

### 5.2. Calculation of Bearing Capacity of Channel Steel Reinforcement Method

(1) Calculation of cracking bearing capacity

The experiment shows that when the tensile zone of concrete reaches the limit tensile strain $\varepsilon_{tu}$, the tensile side of the specimen is provided by concrete and channel steel together to provide tensile stress, and the calculation sketch of the cross-section at cracking is shown in Figure 9.

When cracking of the specimen, the concrete in the tensile zone already has a plastic development zone, and the tensile stress is distributed along the horizontal cross-section in a trapezoidal shape with the maximum stress $f_{t}$. The specific distribution is shown in Figure 9c. However, since the tensile stress in the concrete at the time of cracking is small and when the moment of tensile stress on the neutral layer position is mainly provided by the straight line segment of the tensile stress distribution diagram, the rectangular stress distribution shown by the solid line is used instead of the trapezoidal stress distribution. The compressive stress of concrete is smaller and the compressive stress distribution can be regarded as linear distribution. The maximum compressive stress can be calculated by $\sigma_{cr}=E_{c}\varepsilon_{cr}$, in which $E_{c}$ represents the modulus of elasticity of uncorroded concrete, $\varepsilon_{cr}$ represents the compressive strain at the edge of the concrete away on the axial force side.

Thus, according to the strain geometry relationship shown in Figure 9c, it is obtained that:(2)$$\frac{\varepsilon_{t}{}_{u}}{h-x_{n}}=\frac{\varepsilon_{c}{}_{r}}{x_{n}}=\frac{\varepsilon_{s}{}_{c1}}{h-x_{n}-b_{c}/2}=\frac{\varepsilon_{\mathrm{sc}2}}{h-x_{n}-b_{c}}$$
where: $\varepsilon_{tu}$ is the ultimate tensile strain at the edge of the concrete away from the axial force side; $\varepsilon_{cr}$ is the compressive strain at the edge of the concrete away on the axial force side; $\varepsilon_{sc1}$ is the compressive strain at the edge of the concrete near the axial force side; $\varepsilon_{sc2}$ is the tensile strain at the geometric center of the channel leg; $x_{n}$ is the tensile strain at the waist of the channel; $b_{c}$ is the height of the cross-sectional pressure zone at cracking; *h* is the leg length of the channel, as shown in Figure 10.

In Figure 10, *h* is the cross-sectional thickness of the tunnel lining; *b* is the longitudinal width of the tunnel lining.

Further, according to the balance of forces, it is obtained by, respectively:(3)$$\left\{ \begin{array}{l} N_{\mathrm{cr}}=0.5\;\sigma_{\mathrm{cr}}bx_{n}-f_{t}b\left(h-x_{n}\right)-\sigma_{\mathrm{sc}1}A_{\mathrm{sc}1}-\sigma_{\mathrm{sc}2}A_{\mathrm{sc}2}\\ N_{\mathrm{cr}}\;e=0.5\;\sigma_{\mathrm{cr}}bx_{n}\left(h-\frac{1}{3}x_{n}-\frac{1}{2}b_{c}\right)-f_{t}b\left(h-x_{n}\right)\left(\frac{h-x_{n}}{2}-\frac{1}{2}b_{c}\right)-\frac{1}{2}\sigma_{s2}A_{s2}b_{c} \end{array}\right.$$
where: $\sigma_{sc1}$ is the tensile stress in the leg of the channel; $\sigma_{sc2}$ is the tensile stress in the waist of the channel; $A_{sc1}$ is the cross-sectional area of the leg of the channel; $A_{sc2}$ is the cross-sectional area of the waist of the channel; $A_{sg}$ is the cross-sectional area of the corrugated steel plate.

Jointly (2) and (3) can be obtained as follows.
(4)$$\left\{ \begin{array}{l} N_{\mathrm{cr}}=0.5\;E_{c}\frac{\varepsilon_{\mathrm{tu}}}{h-x_{n}}bx_{n}{}^{2}-f_{t}b\left(h-x_{n}\right)-E_{s}\varepsilon_{\mathrm{tu}}\frac{h-x-b_{c}/2}{h-x_{n}}A_{\mathrm{sc}1}-E_{s}\varepsilon_{\mathrm{tu}}\frac{h-x-b_{c}}{h-x_{n}}A_{\mathrm{sc}2}\\ N_{\mathrm{cr}}e=0.5\;E_{c}\frac{\varepsilon_{\mathrm{tu}}}{h-x_{n}}bx_{n}{}^{2}\left(h-\frac{1}{3}x_{n}-\frac{b_{c}}{2}\right)-f_{t}b\left(h-x_{n}\right)\left(\frac{h-x_{n}}{2}-\frac{b_{c}}{2}\right)-\frac{b_{c}}{2}E_{s}\varepsilon_{\mathrm{tu}}\frac{h-x-b_{c}}{h-x_{n}}A_{\mathrm{sc}2} \end{array}\right.$$
where, $E_{c}$ is the modulus of elasticity of uncorroded concrete; $E_{s}$ is the modulus of elasticity of steel.

Form (4) is a binary equation with *N* and $x_{n}$ can be substituted into the known quantities such as channel steel section parameters to solve for the bearing capacity of the channel steel method of reinforcing specimens when cracking.

(2) Calculation of ultimate bearing capacity

Channel reinforced specimen damage, away from the near axial force side of the tensile area of the concrete has cracked, tensile stress is fully assumed by the channel. However according to the test, the channel has not yielded at this time, the tensile stress can be expressed as $\sigma_{sc}=E_{s}\varepsilon_{sc}$, in which $E_{s}$ represents the modulus of elasticity of steel, $\varepsilon_{sc}$ represents the strain of channel. Thus, the cross-sectional stress distribution of the specimen in the current state is shown in Figure 11b. The edge of the compressive zone reaches the ultimate pressure, the concrete crushes and cracks, and the specimen is completely destroyed.

Similarly, according to the strain geometry relationship and the balance of forces, we can get:(5)$$\left\{ \begin{array}{l} \frac{\varepsilon_{c}{}_{u}}{x_{n}}=\frac{\varepsilon_{s}{}_{c1}}{h-x_{n}-b_{c}/2}=\frac{\varepsilon_{\mathrm{sc}2}}{h-x_{n}-b_{c}}\\ N_{\mathrm{cu}}=\int_{0}^{x_{n}}\sigma_{c}bdx-\sigma_{\mathrm{sc}1}A_{\mathrm{sc}1}-\sigma_{\mathrm{sc}2}A_{\mathrm{sc}2}\\ N_{\mathrm{cu}}e=\int_{0}^{x_{n}}\sigma_{c}bdx\left(h-y_{c}-\frac{b_{c}}{2}\right)-\frac{b_{c}}{2}\sigma_{\mathrm{sc}2}A_{\mathrm{sc}2} \end{array}\right.$$
where, $\varepsilon_{cu}$ is compressive strain of the concrete edge near the axial force side, according to the test to take 250 µε.

By solving the above equation, the calculation formula of section bearing capacity of the component strengthened by channel steel can be obtained, as shown in Equation (6). Then, the ultimate bearing capacity of the specimen reinforcement by channel steel can be obtained by substituting relevant parameters.
(6)$$\left\{ \begin{array}{l} N_{\mathrm{cu}}=0.733f_{c}bx_{n}-\frac{h-x_{n}-b_{c}/2}{x_{n}}E_{s}\varepsilon_{\mathrm{cu}}A_{\mathrm{sc}1}-\frac{h-x_{n}-b_{c}}{x_{n}}E_{s}\varepsilon_{\mathrm{cu}}A_{\mathrm{sc}2}\\ N_{\mathrm{cu}}e=0.733f_{c}bx_{n}\left(h-0.391x_{n}-\frac{b_{c}}{2}\right)-\frac{h-x_{n}-b_{c}}{2x_{n}}b_{c}E_{s}\varepsilon_{\mathrm{cu}}A_{\mathrm{sc}2} \end{array}\right.$$

### 5.3. Calculation of the Bearing Capacity of Corrugated Steel Plate Reinforcement Method

When the corrugated steel plate is used to repair the tunnel lining, it deforms with concrete through the anchor bolt nails, and its damage process also includes two critical states: cracking moment and overall failure moment, and its stress mode is roughly the same as the effect of corrugated steel plate repair, as shown in Figure 12.

Referring to the derivation method in Section 5.2, the formulae for calculating the cross-sectional bearing capacity when the specimen is cracked and damaged as a whole in two limit states after reinforcement by corrugated steel plate method can be obtained in turn, such as Equations (7) and (8).
(7)$$\left\{ \begin{array}{l} N_{\mathrm{cr}}=0.5\;E_{c}\frac{\varepsilon_{\mathrm{tu}}}{h-x_{n}}bx_{n}{}^{2}-f_{t}b\left(h-x_{n}\right)-E_{s}\varepsilon_{\mathrm{tu}}A_{\mathrm{sg}}\\ N_{\mathrm{cr}}e=0.5\;E_{c}\frac{\varepsilon_{\mathrm{tu}}}{h-x_{n}}bx_{n}{}^{2}\left(h-\frac{1}{3}x_{n}\right)-f_{t}b\left(h-x_{n}\right)\frac{\left(h-x_{n}\right)}{2} \end{array}\right.$$
(8)$$\left\{ \begin{array}{l} N_{\mathrm{cu}}=0.733f_{c}bx_{n}-\left(\frac{h}{x_{n}}-1\right)E_{s}\varepsilon_{\mathrm{cu}}A_{\mathrm{sg}}\\ N_{\mathrm{cu}}e=0.733f_{c}bx_{n}\left(h-0.391x_{n}\right) \end{array}\right.$$
where: $\varepsilon_{tu}$ is the ultimate tensile strain of the concrete edge away from the axial force side; $\varepsilon_{sg}$ is the tensile strain of the corrugated steel plate,$\varepsilon_{tu}$= $\varepsilon_{sg}$; $\varepsilon_{cr}$ is the compressive strain of the concrete edge near the axial force side; $x_{n}$ is the height of the pressure zone of the cracked section; $N_{cu}$ is the axial compressive bearing capacity of the cracked section; *e* is the distance from the point of action of the axial force to the concrete joint point of the tension zone,$e=\eta_{s}e_{i}+h/2$; $\sigma_{cr}$ is the compressive stress of the concrete edge near the axial force side; $\sigma_{sg}$ is the tensile stress of the corrugated steel plate; $f_{t}$ is the ultimate tensile strength of the concrete; $A_{sg}$ is the cross-sectional area of the corrugated steel plate.$\varepsilon_{sg}$ is the tensile strain of the corrugated steel plate, $\varepsilon_{tu}$ = $\varepsilon_{sg}$; $\varepsilon_{cu}$ is the compressive strain of the concrete edge near the axial force side; $x_{n}$ is the height of the cracked section compressive zone; $N_{cu}$ is the axial compressive bearing capacity of the specimen when it is completely destroyed; $\sigma_{sg}$ is the tensile stress of the corrugated steel plate; $f_{t}$ is the ultimate tensile strength of the concrete; $A_{sg}$ is the cross-sectional area of the corrugated steel plate.

## 6. Comparative Verification and Engineering Application

In order to compare and analyze the reliability of the simplified design method derived in Section 5, numerical simulations of multiple working conditions are carried out by establishing a finite element model in this section, and the test results, theoretical calculation results and numerical simulation results are compared and analyzed in turn. On this basis, the application is carried out with the dependent project.

### 6.1. Model Establishment

Figure 13 shows the main construction numerical model for the comparative numerical test, in which the channel steel is taken as national standard 8#, 10# and 12#, and the thickness of corrugated steel plate is taken as 3 mm, 4 mm and 5 mm.

In the model, only the elastic working state of steel is considered, and only the ideal elastic material is considered. The isotropic elastic damage model is adopted for concrete materials and is combined with isotropic tensile or compressive plasticity to simulate the inelastic behavior of brittle materials such as concrete, as shown in Figure 14.

### 6.2. Comparative Analysis of Results

In the numerical model, whether the specimen is cracked or not can be judged according to the strain value of the tensile side and the damage factor for the bearing capacity of the component when it is cracked. For the ultimate bearing capacity of the specimen when it was damaged, the maximum axial force value in the loading process was directly read, as shown in Figure 15.

The relevant calculation results are shown in Table 4. By comparison, the results of numerical simulation, experimental testing and theoretical simplification methods are close, with the maximum deviation less than 8%. It can be seen that the calculation method of bearing capacity of corroded components after reinforcement established in Section 5 of this paper is reliable and can be used in the design calculation of corroded lining reinforcement.

### 6.3. HTG Tunnel Rehabilitation Program Design

(1) Program I: channel steel method of reinforcement (corrosion depth of 10 cm)

For the large area sulfate corroded tunnel with corrosion depth of 0–10 cm, channel steel with repair concrete was used to strengthen and repair the corroded parts. The design control section is selected from the section at the minimum value of safety factor at the location of the side wall, the axial force *N* of this section is 531 kN, the bending moment *M* is 78.5 kN·m, which is an eccentric compressive member with large eccentric distance. The length of the element is analyzed along the longitudinal line, that is, the cross-section size of the eccentric compression member is 1 m × 0.45 m. Channel steel type selected GB 18#B, steel type Q345.The channel steel is connected with the existing lining through expansion bolts, and the slightly expansion concrete C30 is poured at the site where the corroded concrete is chiseled. The relevant calculation parameters are shown in Table 5.

After corrosion deterioration of the tunnel lining, the axial force of each part remains almost constant [27], and the bending moment value increases with the increase in corrosion [28], which in turn leads to the continuous increase in the eccentric distance of the lining and the risk of tension cracking on the tensioned side [29]. According to the specification, it is known that the eccentricity distance $e_{0}$ is greater than 0.2 h should be cross-sectional strength review test from the crack resistance point of view. Hence the feasibility test of the repair program is carried out by using Equation (4). The solution is that *M*_cr_ = 360 kN·m. That is, when 18#B channel steel is used to reinforce and repair the lining, the cracking moment of the lining increases to 360 kN, which is much larger than the lining moment value of 78.5 kN·m at this moment, and the lining cross-section retains 3.6 times of safety margin, so it can be seen that the program is feasible.

(2) Program II: corrugated steel plate method (corrosion depth of 20 cm)

For the large area sulfate corroded tunnel with a corrosion depth of 10–20 cm, a corrugated steel plate can be used to reinforce and repair the corroded parts. The design control section selects the section at the minimum safety factor of the side wall position. The axial force N of this section is 527 kN and the bending moment M is 87.5 kN·m, which is an eccentric compressive member with a large eccentricity. The length of the element is analyzed along the longitudinal line, that is, the cross-section size of the eccentric compression member is 1 m × 0.45 m. The corrugated steel plate is made of Q345 steel with a thickness of 5 mm. The steel plate is processed into a corrugated shape by a hydraulic bending machine, and the micro-expanded concrete C30 is poured at the part where the corroded concrete is chiseled. The relevant calculation parameters are as follows: the cross-sectional area of the steel plate *A_sg_* = 0.0067 m^2^; elastic modulus of steel plate *E*_s_ = 2.06 × 10^5^ MPa.

After the tunnel lining is corroded and deteriorated, the axial force of each part is almost constant, and the bending moment value increases with the increase in the corrosion degree, which leads to the continuous increase in the eccentric distance of the lining, and the tensile side faces the risk of being pulled apart and cracked. According to the specification, when the eccentricity $e_{0}$ is greater than 0.2 h, the cross-sectional strength check should be carried out from the anti-crack angle. Therefore, Equation (7) is used to test the feasibility of the repair scheme. The solution is that *M*_cr_ = 458 kN·m. That is, after using a 5 mm thick corrugated steel plate to reinforce and repair the lining, the bending moment of lining cracking increases to 536.5 kN, which is much larger than the bending moment value of lining at this moment of 87.5 kN·m, and the lining section retains 5.1 times of safety margin, and the scheme can play a better role in reinforcing and repairing.

## 7. Conclusions

For the plain concrete lining structure with large area corrosion and deterioration, the existing reinforcement methods cannot meet the needs. The lining concrete has lost structural strength within a certain thickness range, and it is impossible to paste fiber materials or anchor steel plates on the surface of the lost strength during the actual repair, so it is often necessary to chisel out the severely corroded parts before the next step of reinforcement construction. The mechanical transformation mechanism and design method of the reinforced and repaired lining structure is not complete. In the actual construction, the repair design scheme is mostly determined by the method of empirical judgment. The reliability and economy of the design lack of rigorous theoretical basis. From this investigation, the following findings may be drawn:
(1)Based on the HTG tunnel project, the basic distribution pattern of HTG tunnel lining engineering diseases and macro mechanical properties of corroded concrete were mastered. The results show that: HTG tunnel is located in a severe chemical erosion environment, the measured sulfur dioxide concentration is as high as 618.9 mg/L. Affected by the erosion environment, the tunnel lining surface concrete is swollen and collapsed, with serious loss of strength, the measured concrete compressive strength is 17–24 MPa, which has influenced the overall stability of the lining structure.(2)According to the site survey and the existing engineering experience, the corrugated steel plate or channel steel method reinforcement plan was designed and indoor tests were conducted, respectively. The test results showed that the damage process of the specimens reinforced on the tensile side could be divided into the uncracked stage and cracked working stage, and the cracking load and damage load of the specimens were significantly improved. Compared with the unreinforced plain concrete specimens, the tensile stiffness of the members could be significantly improved by using the channel steel method or the corrugated steel plate method on the tensile side.(3)After analyzing the test results of the reinforced specimen, the stress characteristics and damage process of the reinforced specimen are obtained. The bearing capacity of the reinforced specimen is divided into the cracking limit bearing capacity and the limit bearing capacity at the time of damage, and the bearing capacity calculation methods of the channel reinforcement method and the corrugated steel plate reinforcement method can be deduced. The results of comparison and analysis show that: the results of numerical simulation, experimental test and theoretical simplification method are close to each other, and the maximum deviation is less than 8%. The established method of calculating the bearing capacity after reinforcement of corroded members is reliable and can be used in the design calculation of corroded lining reinforcement.

## Figures and Tables

**Figure 1 materials-16-02871-f001:**
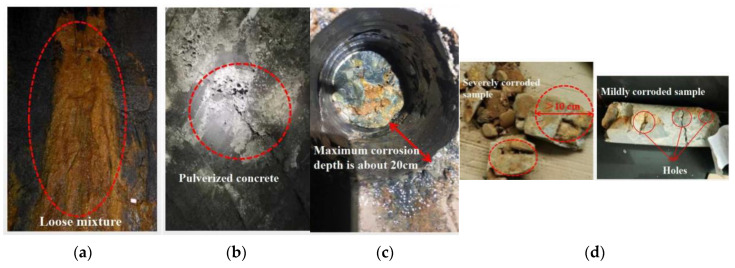
Investigation of HTG tunnel lining corrosion. (**a**) Salt attachment; (**b**) Pulverized surface concrete; (**c**) Measured corrosion depth; (**d**) Drill core sample.

**Figure 2 materials-16-02871-f002:**
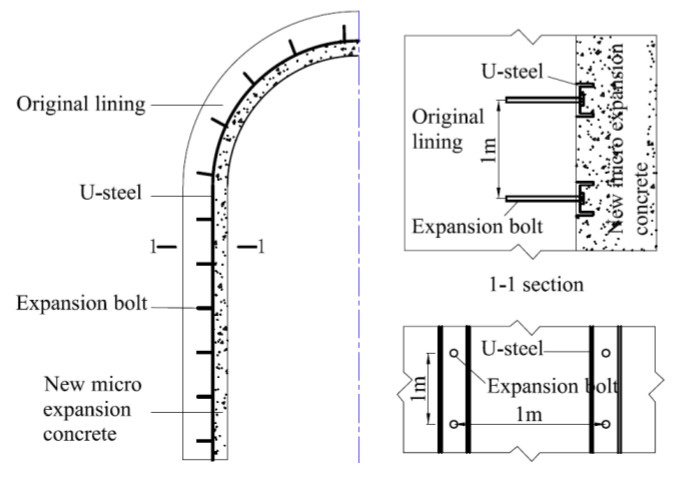
Channel steel reinforcement method.

**Figure 3 materials-16-02871-f003:**
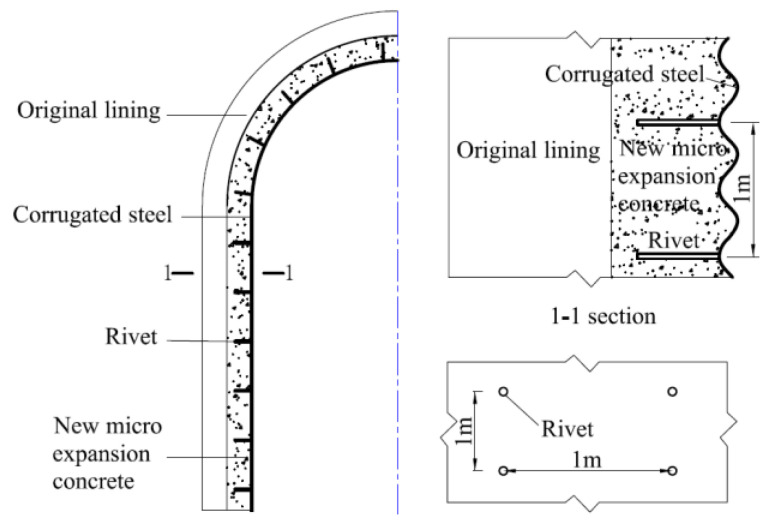
Corrugated steel plate reinforcement method.

**Figure 4 materials-16-02871-f004:**
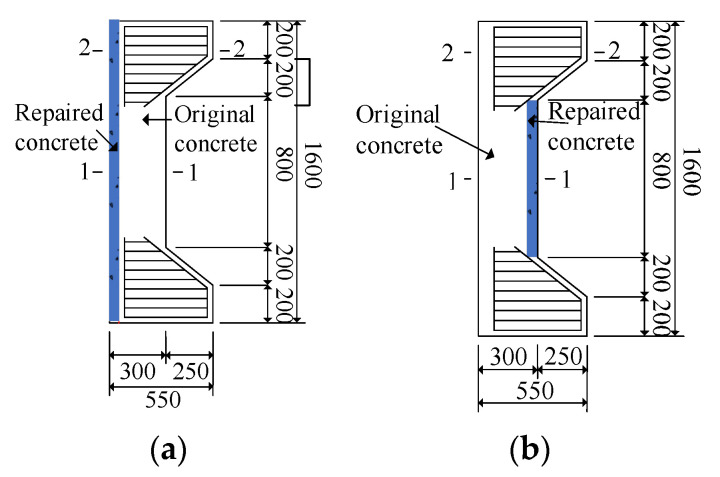
Design of reinforced specimen (Unit: mm). (**a**) Tensile side reinforcement specimen; (**b**) Compression side reinforcement specimen.

**Figure 5 materials-16-02871-f005:**
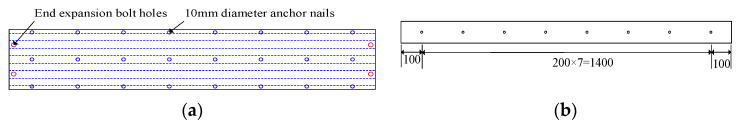
Steel plate processing diagram. (**a**) Distribution of anchor bolts and bolt holes; (**b**) Distribution of slotted steel bolt holes; (**c**) Corrugated steel plate cross-sectional drawing; (**d**) Cross-sectional view of channel steel.

**Figure 6 materials-16-02871-f006:**
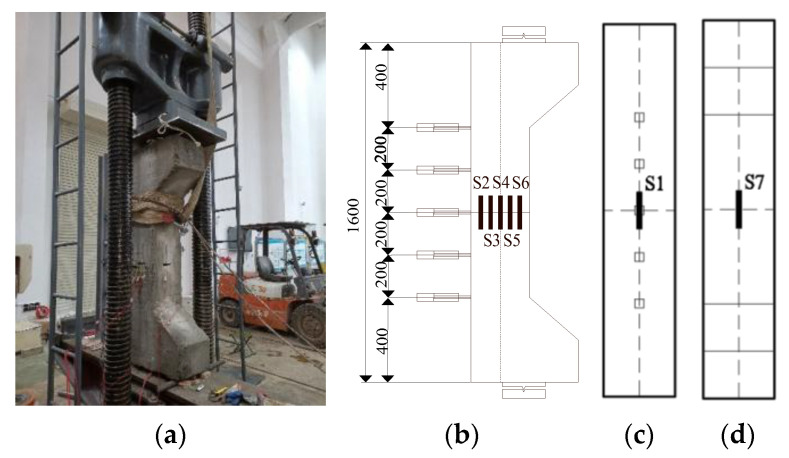
Specimen loading and measurement point arrangement diagram. (**a**) Loading map; (**b**) Side; (**c**) Pulled side; (**d**) Pressurized side.

**Figure 7 materials-16-02871-f007:**
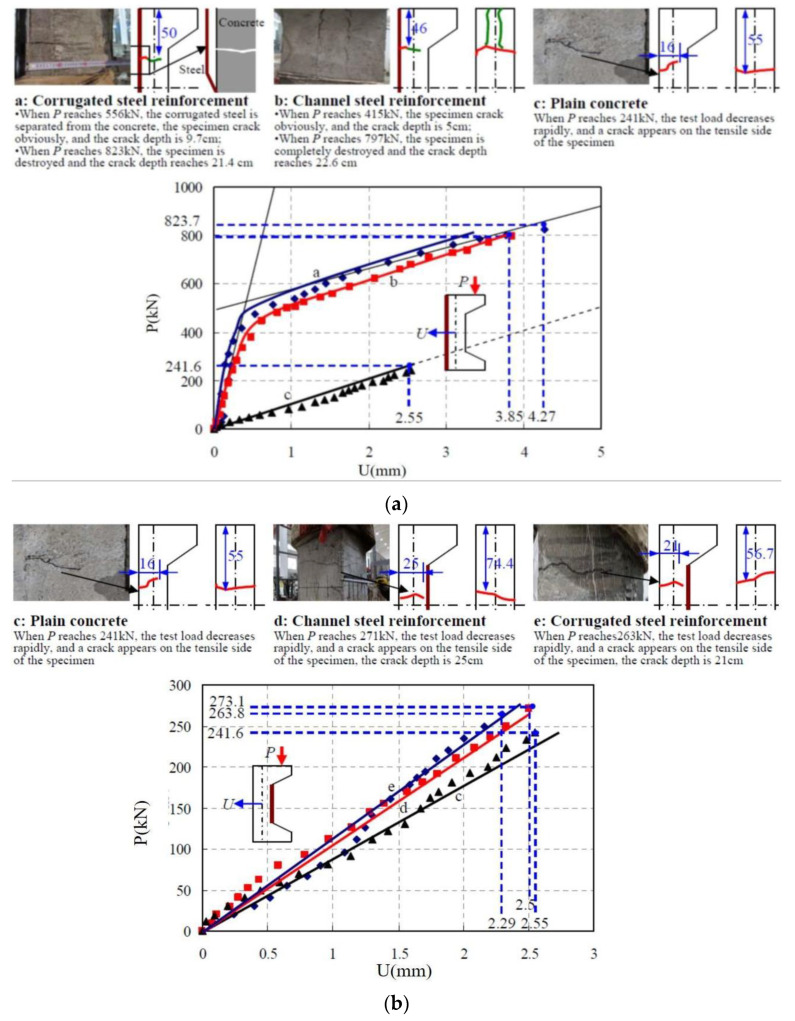
Load–deflection test results for each specimen. (**a**) Tension side reinforcement; (**b**) Compression side reinforcement.

**Figure 8 materials-16-02871-f008:**
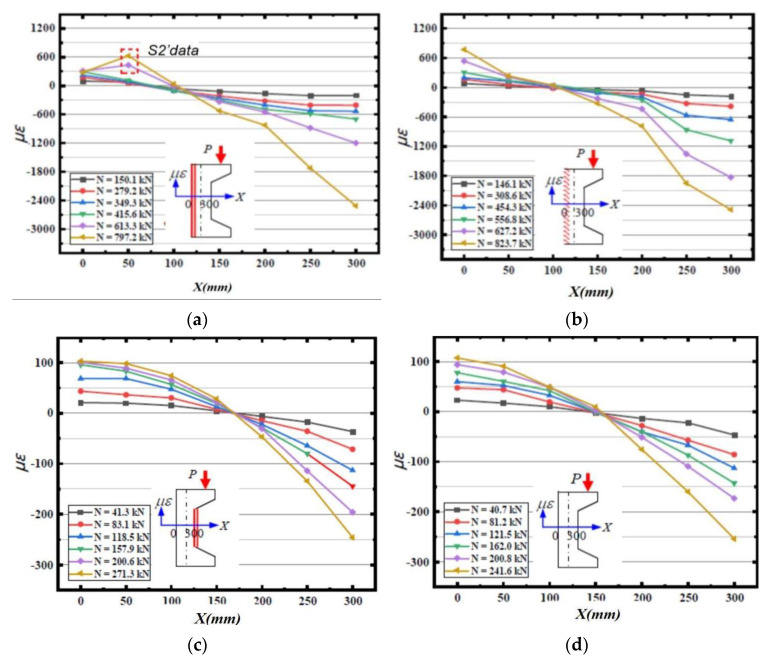
Strain distribution test results of each specimen section. (**a**) Tensioned side channel reinforcement method; (**b**) Tensile side corrugated steel plate reinforcement method; (**c**) Steel channel reinforcement on pressurized side; (**d**) Corrugated steel plate reinforcement on the pressurized side; (**e**) Unreinforced plain concrete.

**Figure 9 materials-16-02871-f009:**
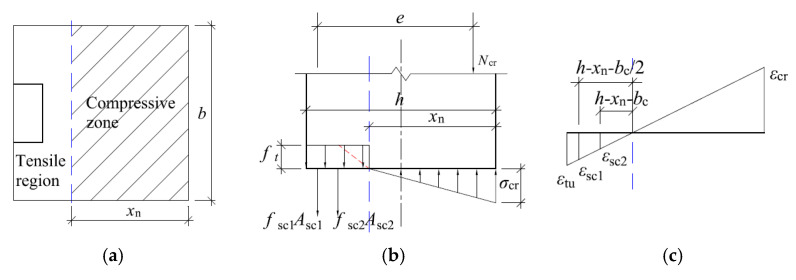
Sketch of calculation when the specimen is cracked by channel reinforcement method. (**a**) Cross-sectional view at cracking; (**b**) Cross-sectional stress distribution at cracking; (**c**) Cross-sectional strain distribution at cracking.

**Figure 10 materials-16-02871-f010:**
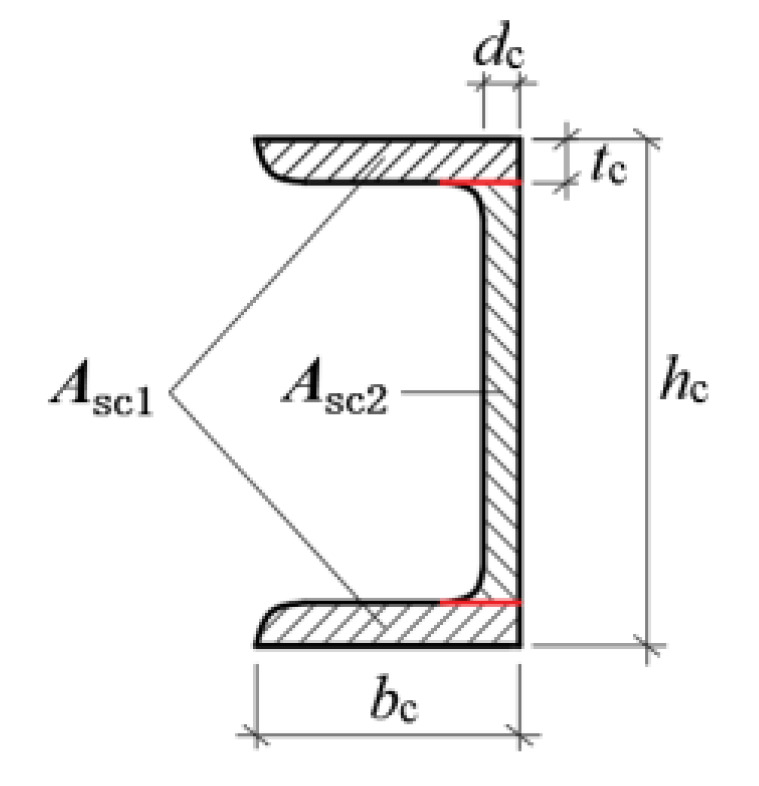
Cross-sectional drawing of channel.

**Figure 11 materials-16-02871-f011:**
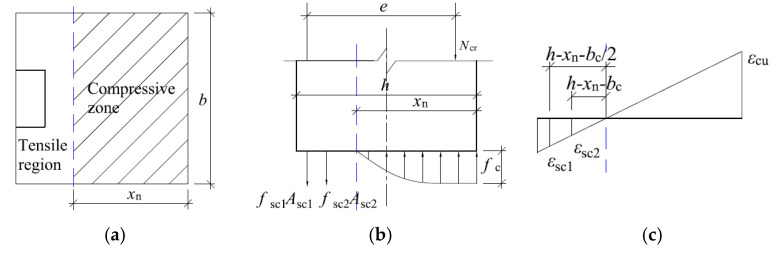
Sketch of the calculation when the specimen is damaged by the channel reinforcement method. (**a**) Cross-sectional view during crush damage; (**b**) Cross-sectional stress distribution during crush damage; (**c**) Cross-sectional strain distribution during crush damage.

**Figure 12 materials-16-02871-f012:**
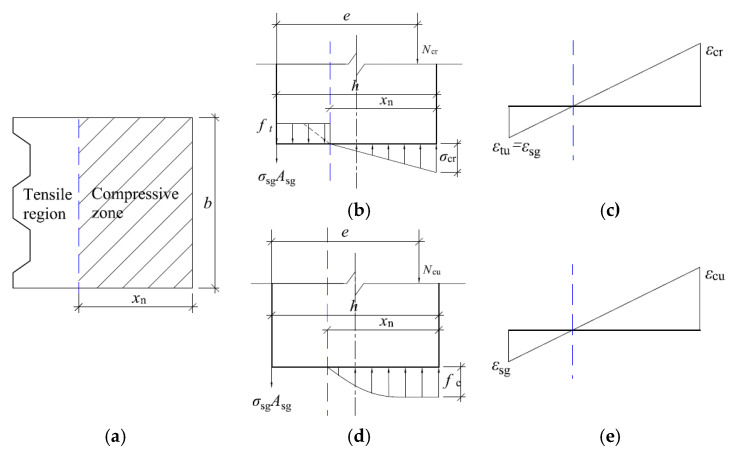
Sketch of bearing capacity calculation of specimen by corrugated steel plate reinforcement method. (**a**) Cross-sectional view; (**b**) Cross-sectional stress distribution at cracking; (**c**) Cross-sectional strain distribution at cracking; (**d**) Cross-sectional stress distribution during crush damage; (**e**) Cross-sectional strain distribution during crush damage.

**Figure 13 materials-16-02871-f013:**
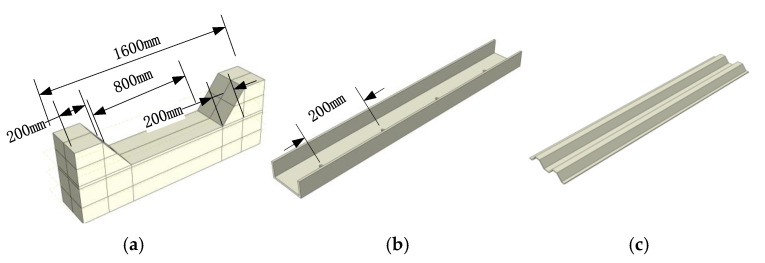
Numerical model. (**a**) Main part of bias column; (**b**) Channel steel; (**c**) Corrugated Steel Sheets.

**Figure 14 materials-16-02871-f014:**
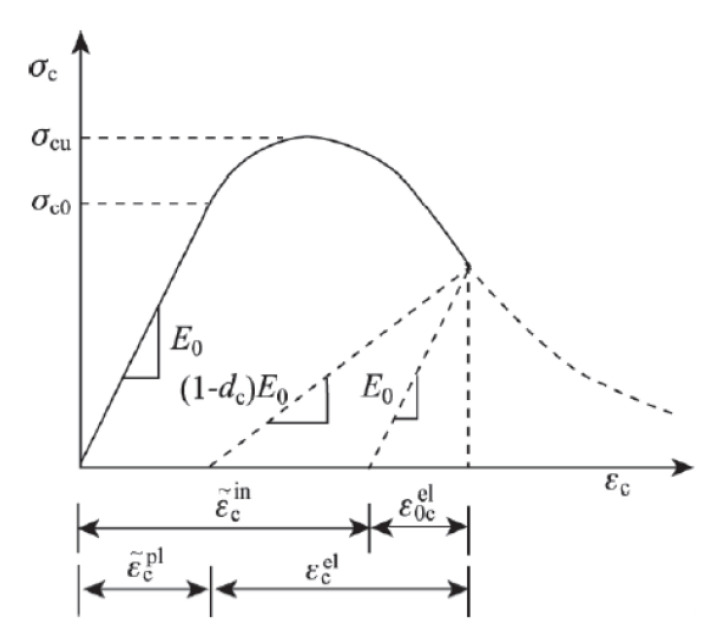
Concrete damage plasticity model.

**Figure 15 materials-16-02871-f015:**
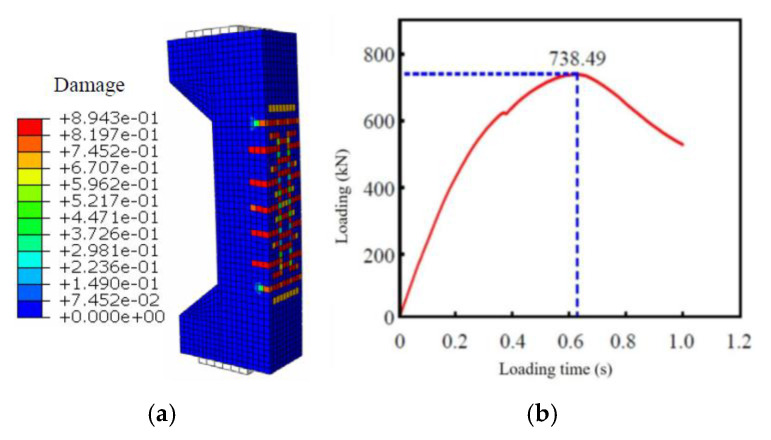
Numerical test results of typical specimens. (**a**) Model cracking under tension; (**b**)Ultimate load carrying capacity at model destruction.

**Table 1 materials-16-02871-t001:** Specimen design specification table.

Specimen Label	Channel Steel Reinforcement Method	Corrugated Steel Plate Reinforcement Method	Plain Concrete
CGZ-01	CGZ-02	GBZ-01	GBZ-02	SZ-01
Specimen height/mm	1600
Cross-section after picking groove/mm	300 × 200	/
Cross-section after repair/mm	300 × 300	300 × 325	/
Eccentric distance/mm	270
Plate type	10#plain channel steel	4 mm thick Q345 corrugated steel	/
Reinforcement position	tensile side	compression side	tensile side	compression side	/
1-1 Cross-section detail	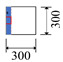	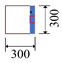	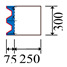	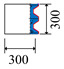	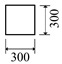
2-2 Cross-section detail(supporting Bracket part)	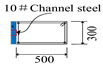	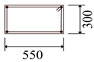	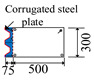	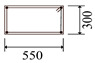	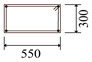

**Table 2 materials-16-02871-t002:** The repaired concrete ratio.

Materials	Cement	Coarse Aggregate	Fine Aggregate	Water	Water-Reducing Agent	Expanding Agent
weight	484.32	753.11	862.69	208.26	1.94	2.85

**Table 3 materials-16-02871-t003:** Flexural stiffness of specimen.

Repair Location	Repair Method	Carrying Capacity *N*_u_/kN	Deflection *f*/mm	Flexural Stiffness *EI*/kN·m^2^
/	Unreinforced plain concrete	241.6	2.55	2949.0
Pulled side	Channel method	797.2	3.85	6445.1
	Corrugated steel sheet method	823.7	4.27	6003.6
Pressurized side	Channel method	271.3	2.29	3687.5
	Corrugated steel sheet method	263.8	2.5	3284.4

**Table 4 materials-16-02871-t004:** Comparison of calculation results.

Reinforcement Method	Model or Thickness	Cracking Load(kN)			Deviation/%	Cracking Load(kN)			Deviation/%
		Simplify Calculations	Numerical Simulation	Test Results		Simplify Calculations	Numerical Simulation	Test Results	
Channel steel	8#	397.1	376.1	\	5.3	736.2	715.3	\	2.8
	10#	432.4	400.6	415.6	7.4	745.4	738.5	797.2	7.4
	12#	467.6	486.9	\	4.0	754.6	743.7	\	1.4
Corrugated Steel plate	3 mm	496.3	482.6	\	2.8	802.3	795.2	\	0.9
	4 mm	554.5	531.4	556.8	4.6	833.8	801.6	823.7	3.9
	5 mm	597.1	584.3	\	2.1	857.1	832.7	\	2.8

**Table 5 materials-16-02871-t005:** Calculation parameters of reinforcement scheme by channel steel method.

Channel Steel Specifications	Modulus of Elasticity of Channel Steel *E*_s_ (MPa)	Height *h* (m)	Leg Width *b* (m)	Waist Thick *d* (m)	Channel Cross-Sectional Area *A*_sc1_ (m^2^)	Channel Cross-Sectional Area *A*_sc2_ (m^2^)
18#B	2.06 × 10^5^	0.18	0.007	0.009	0.0015	0.0011
Modulus of elasticity of concrete *E*_c_ (MPa)		Tensile ultimate strength of concrete *f*_t_ (MPa)			Section width of the member *b*(m)	Section height of members *h*(m)
2.98 × 10^4^		2.1			1	0.45

## Data Availability

Data are contained within the article.

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
