# Peer review of "Novel Reinforcing Techniques and Bearing Capacity Analysis for Tunnel Lining Structures with Extensive Corrosion"

_materials, 2023, doi:10.3390/ma16072871_

Round 1

Reviewer 1 Report

The reviewer appreciates the effort executed by the authors for producing this work. However, the reviewer has some comments, which may upgrade the quality of paper as follows:

What is the novelty in your study to write “novel” in the title? and please, illustrate that in the abstract and introduction

In Abstract

1-From the point of geologic view, the reviewer thinks that there is not a term called erosive stratigraphic environment, but it is called erosive environment without “stratigraphic”. The stratigraphy is the branch of geology concerned with the order and relative position of strata and their relationship to the geological timescale.

2-“disease issues” is not suitable

3-Corrosion concrete macro-mechanical properties, I think there is a comma.

4-Please, define HTG abbreviation

Results and discussion

Did you differentiate between the efficiency of each of the channel steel method or corrugated steel plate method? Please illustrate that in the results and discussion and conclusion and abstract sections, with focusing on the most qualified and appropriate.

The compressive strength 17-24 MPa, this strength range is within the structural concrete conditions according to requirements of ACI 318–14, which have the minimum conditions of structural concrete to be 17.20 MPa. I mean do you think that this tunnel strength needs reinforcement.

In Figure 13, put the the unit of mm within the figure not in the caption.

In all figures like 7 and 8, and 15, Please, the units in brackets not after dash

Conclusion
Do you think that a word of disease is suitable for using in Engineering? I think you should change it.

Author Response

1.In Abstract

1a-From the point of geologic view, the reviewer thinks that there is not a term called erosive stratigraphic environment, but it is called erosive environment without “stratigraphic”. The stratigraphy is the branch of geology concerned with the order and relative position of strata and their relationship to the geological timescale.

1b-“disease issues” is not suitable

1c-Corrosion concrete macro-mechanical properties, I think there is a comma.

1d-Please, define HTG abbreviation

Response:

Thank you for your rigorous advice.

  • We have reviewed the full text and modified the corresponding expression “erosive environment” according to the suggestions of the reviewer. (Line 14.)
  • The author has checked again through the academic database, the current common expression of engineering structure disease is "disease", in order to distinguish from biological disease, the authors added the restriction condition, changed to "engineering disease".  (Lines 15,18,91,465.)
  • The sentence you mentioned has been modified to "Based on HTG tunnel project, the basic distribution rule of tunnel lining corrosion and macro mechanical properties of corroded concrete were explored in this paper through engineering disease site investigation." Here is our oversight. Your suggestion of adding commas is valuable, but we have corrected the expression of the sentence by changing the word order. Although we may not adopt your suggestions for modification, we would still appreciate your constructive comments. (Lines 16-19.)
  • The "HTG" used in this paper is the abbreviation of HuangTuGang tunnel name.  Due to confidentiality requirements, the abbreviation is used in the manuscript. (Line 16.)

2.Results and discussion

2a. Did you differentiate between the efficiency of each of the channel steel method or corrugated steel plate method? Please illustrate that in the results and discussion and conclusion and abstract sections, with focusing on the most qualified and appropriate.

2b. The compressive strength 17-24 MPa, this strength range is within the structural concrete conditions according to requirements of ACI 318–14, which have the minimum conditions of structural concrete to be 17.20 MPa. I mean do you think that this tunnel strength needs reinforcement.

2c. In Figure 13, put the unit of mm within the figure not in the caption.

2d. In all figures like 7 and 8, and 15, Please, the units in brackets not after dash.

Response:

1)Thank you for your review. The original intention of this paper is to find a reasonable reinforcement disposal method for the corroded tunnel lining structure. The focus of this study is the design and feasibility of the reinforcement scheme. Therefore, the application scope of the two reinforcement schemes is only distinguished in the actual study. As explained in the paper, channel steel method is applicable to the corrosion depth of 0-10cm, while corrugated steel plate method is applicable to the corrosion depth of more than 10cm, and the reinforcement efficiency of the two schemes is not compared and analyzed. Of course, whether from engineering practice experience or intuitive theoretical understanding, corrugated steel plate method of reinforcement ability is better, but the specific data comparison, the author has not carried out the corresponding demonstration. The analysis of the efficiency of the two reinforcement methods may be used as a train of thought for further study.

2) Thank you for your review. As a kind of superstatic structural system, the current structural design of tunnel is very close to the idea of "reasonable arch axis", so simply from the structural bearing capacity, 17.20MPa can basically meet the requirements. However, the tunnel works serve for the passage of trains. From the perspective of its normal performance, the strength of 17.20MPa and its corresponding impermeability and other performance will not meet the requirements of the safe operation of the line. Therefore, it is clearly required in the actual engineering and needs to be reinforced. Corresponding to the current code in China, regarding the strength of tunnel lining concrete, it is clearly required to reach C35 grade from the consideration of its durability.

3) Thank you for your review. We have modified Figure 13 according to your suggestion.

4) Thank you for your review. The units of Figures 7,8 and 15 have been in brakes and not after dash.

  1. Conclusion
    Do you think that a word of disease is suitable for using in Engineering? I think you should change it.

Response:

Thank you for your review. We have verified again through the academic database, and the current common expression of engineering structure disease is "disease". To distinguish the biology disease, we added a defining word, namely “engineering disease”.  (Lines 15,18,91,465.)

Reviewer 2 Report

Regarding the work "New reinforcement techniques and bearing capacity analysis for tunnel lining structures with extensive corrosion".

The following should be considered:

• The summary can be improved.

• The bibliography must be updated because there is little updated bibliography.

• Figures and tables should be improved and be in an appropriate and legible format. For example, in figure 8 to S2 it does not put the symbols in English

Regarding the objective of the work, it is worth noting; as well as the working hypothesis. On the other hand, emphasis should be given to the innovation of the work and its specific contribution, and a discussion section should be added, and the conclusions should be improved.

Normally when it comes to stratigraphy, the horizons of geological interest are studied in greater depth; It is important the geological sections (stratigraphic - structural), structural planes with integration of lithofacies of interest, at levels of chronostratigraphic horizons, also in some cases it is necessary to establish by means of DRX the phases formed to accurately establish the formation. of the minerals and their possible passivation with another mineral.

Author Response

1.The summary can be improved.

Response:

We have added the innovation of the work and its specific contribution in the first paragraph of conclusion. Hence, the conclusion part has been improved.

2.The bibliography must be updated because there is little updated bibliography.

Response:

Thank you for your valuable suggestion! We have searched for many lasted and relevant articles and cited them as references.

3.Figures and tables should be improved and be in an appropriate and legible format. For example, in figure 8 to S2 it does not put the symbols in English

Response:

Some improvements have been made to the pictures and tables, such as adjusting the position of the units and translating the notes in Figure 8 into English. Thank you for your valuable comments!

4.Regarding the objective of the work, it is worth noting; as well as the working hypothesis. On the other hand, emphasis should be given to the innovation of the work and its specific contribution, and a discussion section should be added, and the conclusions should be improved.

Response:

The working objective of section 5.1 is to simplify the calculation process of large area corrosion lining reinforcement design method. The working hypothesis is actually section 5.1. We have added the innovation of the work and its specific contribution in the first paragraph of conclusion and hence we didn’t add the discussion section. Nevertheless, we are still grateful to your valuable suggestions very much.

5.Normally when it comes to stratigraphy, the horizons of geological interest are studied in greater depth; It is important the geological sections (stratigraphic - structural), structural planes with integration of lithofacies of interest, at levels of chronostratigraphic horizons, also in some cases it is necessary to establish by means of DRX the phases formed to accurately establish the formation. of the minerals and their possible passivation with another mineral.

Response:

Thank you for your helpful suggestions! However, the focus of this study is only on the restoration and reinforcement of tunnel structures in the stratum containing erosive salts, and geomechanics is not involved. In addition, we have not found any papers that are very relevant to the stratigraphy studies you mentioned, so we cannot cite them in our article. Nonetheless, thank you very much for your valuable comments.

Reviewer 3 Report

Review report: Novel reinforcing techniques and bearing capacity analysis for tunnel lining structures with extensive corrosion

The abstract part should be written in the spirit of scientific expression. The subject is interesting in the aspect of material science which is directly integrated into engineering problems that should be resolved. But the Material is a scientific journal, so the text should be prepared according to Journal rules.

The Figures are well presented in the text.

The results need to be discussed with literature data. Authors should add literature in paragraphs 219-225, and 229-248.

Lines 249-252. The sentence is in conclusion or in Abstract form. The main goals of the paper should be written in the introduction part. So lines 249-252 should be rewritten in this section, and the main goals of the study are mandatory to be written in the introduction part.

Lines 387-395: References is needed to be added in the text regarding sulfate corrosion behavior.

Please delete the word Main in the conclusions. The conclusion part is the conclusion part. The scientific importance and application of this research also should be emphasized in the conclusion. 

Author Response

1.The abstract part should be written in the spirit of scientific expression. The subject is interesting in the aspect of material science which is directly integrated into engineering problems that should be resolved. But the Material is a scientific journal, so the text should be prepared according to Journal rules.

Response:

Thank you for your review. We have rearranged the abstract and modified the excessively long sentences into several short sentences. In addition, the full text has been revised in accordance with the specification of Materials journal.

2.The Figures are well presented in the text.

Response:

Thank you for your positive comment!

3.The results need to be discussed with literature data. Authors should add literature in paragraphs 219-225, and 229-248.

Response:

Thank you for your review. We have added the specific experimental data in the paragraph. (Lines 271-277 and Lines 282-290.) We have added the following literature in the location you mentioned:

[20] Tiberti, G.; Minelli, F.; Plizzari, G. Reinforcement optimization of fiber reinforced concrete linings for conventional tunnels. Compos. Part B-Eng., 2014, 58, 199-207. https://DOI:10.1016/j.compositesb.2013.10.012.

[30] Lei, M.F. A FDEM approach to study mechanical and fracturing responses of geo-materials with high inclusion contents using a novel reconstruction strategy. Eng. Fract. Mech., 2023, 282, 109171. 

[31] Gong, C.J.; Kang, L.; Zhou, W.H.; Liu, L.H.; Lei, M.F. Tensile performance test research of hybrid steel fiber reinforced self-compacting concrete. Mat., 2023, 16, 1114.

4.Lines 249-252. The sentence is in conclusion or in Abstract form. The main goals of the paper should be written in the introduction part. So lines 249-252 should be rewritten in this section, and the main goals of the study are mandatory to be written in the introduction part.

Response:

We have added quantitative expression to this paragraph you mentioned, thus changing it from the expression of conclusion to the expression of analysis. The main goals of the study have been written in the introduction part.

5.Lines 387-395: References is needed to be added in the text regarding sulfate corrosion behavior.

Response:

Thank you for your review. We have added the following literature in the location you mentioned:

[27] Yu C. Deterioration process and mechanism of cement - based materials under sulfate erosion. Ph.D. Thesis. Southeast University, Nanjing, China, 2013. 

[28] Lei M.F.; Peng L.M.; Shi C.H. Experimental study on evolution law of mechanical properties of tunnel structure suffering ambient sulfate. China Civil Engineering Journal, 2013, 46, 126-132. 

[29] Duan L.; Li Y.H.; Wu J.H. Stability evaluation for the lining structures of tunnels with large corrosion areas in sulfate environment. Modern Tunnelling Technology, 2022, 59, 212-220. Lin, Y.X., Ma, J.J., Lai, Z.S.; Huang, L.C.;

[30] Lei, M.F. A FDEM approach to study mechanical and fracturing responses of geo-materials with high inclusion contents using a novel reconstruction strategy. Eng. Fract. Mech., 2023, 282, 109171. 

[31] Gong, C.J.; Kang, L.; Zhou, W.H.; Liu, L.H.; Lei, M.F. Tensile performance test research of hybrid steel fiber reinforced self-compacting concrete. Mat., 2023, 16, 1114.

Round 2

Reviewer 1 Report

Accepted